# Seasonal Variation in Stable Isotope Ratios of Cow Milk in Vilnius Region, Lithuania

**DOI:** 10.3390/ani9030069

**Published:** 2019-02-26

**Authors:** Andrius Garbaras, Raminta Skipitytė, Justina Šapolaitė, Žilvinas Ežerinskis, Vidmantas Remeikis

**Affiliations:** Mass Spectrometry Laboratory, Center for Physical Sciences and Technology, LT-10257 Vilnius, Lithuania; raminta.skipityte@ftmc.lt (R.S.); justina.sapolaite@ftmc.lt (J.Š.); zilvinas.ezerinskis@ftmc.lt (Ž.E.); vidmantas.remeikis@ftmc.lt (V.R.)

**Keywords:** stable isotopes, seasonal variation, cow milk

## Abstract

**Simple Summary:**

The composition of milk is fundamentally dependent on the feeding of the cows, and thereby on a particular environment. However, stable isotope composition is not always constant in the environment and can even change in the same area. Thus, the aim of this study was to better understand the impact of water and food sources on oxygen, carbon, and nitrogen stable isotope ratio (^18^O/^16^O, ^13^C/^12^C, and ^15^N/^14^N) distribution and fractionation in cow milk in different seasons of the year. Cow milk, drinking water, feed, and precipitation water were collected for three years from the same location. Overall, combined δ^18^O, δ^13^C, and δ^15^N analysis of cow milk provides valuable information about the amount of variation in stable isotope ratios in natural samples. This study can be helpful for studies of milk product authentication.

**Abstract:**

Various studies have shown that stable isotope analysis has the potential to verify the geographic origin of foods and drinks. However, stable isotope composition is not always constant in the environment and can even change in the same area. Dairy products are of particular interest as a group of foods that play an important role in feeding the population. The composition of milk is fundamentally dependent on the feeding of the cows, and thereby on a particular environment. To better understand the amount of variation in δ^18^O, δ^13^C, and δ^15^N values in the milk from the same area, we measured stable isotope ratios in cow milk water, artesian water, and precipitation (δ^18^O) as well as in bulk milk samples (δ^13^C and δ^15^N) collected in 2014–2016. Different water and food sources were available during the winter (artesian water only and dry grass) and summer (artesian water and fresh grass), and spring and autumn seasons reflected transitional periods. Oxygen stable isotope ratios in milk water were relatively lower in winter and transitional seasons and higher in summer, showing the dependence on the main water source. δ^13^C values reflected particular food sources. This study shows the applicability of the stable isotope ratio method in linking cow milk to specific environments and reveals the amount of variation in stable isotope ratios in the same area. These results could be valuable for other studies on geographical origin determination of dairy products.

## 1. Introduction

Increasing demand for authentic and high-quality foods and drinks requires analytical tools to solve the complex problems related to food production, quality assessment, environmental control, and related issues. The authenticity of dairy products is becoming a subject of increasing interest forscientists, consumers, producers, and policymakers, especially as the price of some products can be higher than their alternatives. The ability to differentiate various dairy products from different countries or regions has value for ensuring fair competition between producers and protecting consumers against misleading due to mislabeling fraud [1]. A range of analytical methods have been developed to address this issue. The stable isotope ratio method has already shown its applicability to solve the authenticity problems of milk products [2,3,4,5,6,7,8], beef [9,10,11], olive oil [12,13], honey [14,15], wine [16], and juices [17]. Ehtesham et al. [18] showed that the δ^2^H and δ^13^C of four fatty acids and bulk milk powder were found to be correlated with regional production area in New Zealand. In the United States, the δ^2^H and δ^18^O values of paired milk and cow drinking water were related, suggesting the potential for geographical origin assignment using stable isotope analysis [19]. Liu et al. [20] found that δ^18^O values in goat milk water were identical to that in corresponding drinking water in China, while Garbaras et al. [21] observed variation of δ^18^O values in the cow milk water depending on the region in Belarus.

Stable isotopes fractionate due to chemical and physical factors and thus form different isotopic landscapes. In nature, the stable isotope (SI) ratio varies from one trophic level to another. Consumers have a 3–5‰ higher stable isotope ratio of nitrogen (expressed as δ^15^N), on average, compared to dietary sources, and in the case of carbon stable isotope ratios, this fractionation is lower and reaches ~1–2‰ [22]. Thus, the stable isotope ratio of nitrogen is usually used to differentiate between trophic levels and δ^13^C is usually used to determine the main food source of the animals. For example, the stable isotope ratios of carbon of cattle fed on maize can help to distinguish from cattle fed on grass, as plants have evolved different photosynthetic pathways and are known as C_3_ (most temperate crops, wheat, beans), C_4_ (corn, sugarcane), and CAM (Crassulacean acid metabolism, e.g., pineapple) plants [23].

Specific local conditions can alter the stable isotope baseline, which, in turn, can be transferred to different parts of food webs [24,25]. Therefore, understanding the dynamics of specific environments is essential. The stable isotope ratios of oxygen in local water are mainly influenced by precipitation [26], which can also be estimated using isotope maps (see European δ^18^O map [27]). Manca et al. [28] demonstrated the potential of three stable isotope ratios (^13^C/^12^C, ^34^S/^32^S, and ^18^O/^16^O) to segregate local cheese from other cheeses of non-local origin. However, some isotopic ratios alone (^15^N/^14^N and ^34^S/^32^S) are weaker in milk origin assessment, but might be useful in combination with other indicators [29]. 

There are three main sources of oxygen in animals, including humans: atmospheric oxygen, water, and food. According to Daux et al. [30], for many organisms, water is essential and is usually derived from direct sources such as rivers, lakes, or rainwater. Plants or other food resources are only a secondary source of water. Part of the water is removed from the body, and there is a certain balance between the ratio of the source water isotopes and the water absorbed by the individuals. Thus, the δ^18^O of the body’s tissues reflects the source δ^18^O at a ratio of approximately 1:1 [31]. Therefore, oxygen stable isotope ratios are commonly used for regional origin assignment, and carbon and nitrogen stable isotope ratios are used for determining the feeding practices of growing animals [21,32]. 

The aim of this study was to better understand the impact of water and food sources on δ^18^O, δ^13^C, and δ^15^N distribution and fractionation in cow milk in different seasons of the year. Oxygen stable isotope values in raw milk were compared with main drinking water sources of the exact seasons. These results can be valuable for other studies on milk authentication.

## 2. Materials and Methods

### 2.1. Materials

In this study, milk, drinking water, precipitation, and grass were collected repeatedly from 2014 to 2016. More than 150 samples were collected in this period. A dairy cow owned by a local countryside farmer in the Vilnius region, Lithuania (Figure 1) was selected for milk sample collection. The cow was housed indoors during the winter period, fed dry hay, and watered with local artesian water. During the summer season, it was kept outdoors and fed with fresh grass and watered with artesian water. Milk production was 20–30 L/day according to the season.

Raw milk and artesian water samples were frozen immediately after the collection to avoid evaporation and microbial activity that can lead to possible fractionation. Precipitation samples were collected in a water vessel at particular time periods. Care was taken during hot periods when the evaporation increases, and samples were collected as soon as possible after precipitation. All the collected samples were labeled and kept in different vials in a freezer at −18 °C. We used 50-mL vials for sample collection and storage. The samples were defrosted just before sample preparation and the stable isotope ratio measurements.

### 2.2. Sample Pretreatment

We centrifuged 1.5 mL of defrosted milk at 20,000× *g* for 30 min at room temperature. The upper layer was removed. If necessary, the centrifugation step was repeated in order to remove all residual material. The supernatant was transferred into a vial and later used for the oxygen stable isotope analysis. The scheme of the purification process for the milk samples is described in Scampicchio et al. [33]. Precipitation and artesian water samples were analyzed just after melting; we used 500-μL samples for either water or precipitation sample measurement.

For carbon and nitrogen stable isotope ratio analysis, bulk milk samples were freeze-dried before measurements and then homogenized. 

### 2.3. Ethical Approval

Ethics approval was not required for this paper/research.

### 2.4. Isotope Ratio Mass Spectrometry

For δ^18^O measurements, 500 μL of milk water sample was put into an open vial using a disposable syringe and sealed with new septa. Residual air in the vials was removed from the sample vials following an automated autosampler-assisted flushing procedure, which uses a mixture of 0.2% up to 1% CO_2_ in He. Measurements were carried out after an equilibration time of 24 h at 24 °C. The sampling loop aliquoted 100 μL samples of the headspace into an isothermal gas chromatograph, where CO_2_ was separated from any other gas species. The use of repetitive loop injection (1–2 min per replicate) allowed the precision to approach comparable levels to that of a dual inlet system. Measurements were performed with Gas Bench II (Thermo, Bremen, Germany) coupled to a Thermo V Advantage (Thermo, Bremen, Germany) Isotope Ratio Mass Spectrometer. 

For δ^13^C and δ^15^N measurements, freeze-dried milk samples were weighted into the tin capsules and measured with an Elemental Analyser Flash EA1112 (Thermo, Bremen, Germany) linked to a Thermo V Advantage (Thermo, Bremen, Germany) Isotope Ratio Mass Spectrometer.

Stable isotope data are reported as δX values (where X represents the heavier isotope ^18^O, ^15^N, or ^13^C). Delta values are dimensionless quantities that represent the difference in isotope ratio of a sample relative to an internationally agreed zero-point (defined by the International Atomic Energy Agency (IAEA)) as shown in Equation (1):δX = [R_sample_/R_RM_ − 1](1)
where R_sample_ is ^18^O/^16^O, ^15^N/^14^N, and ^13^C/^12^C of the sample and R_RM_ is ^18^O/^16^O, ^15^N/^14^N, and ^13^C/^12^C of the scale zero-point. Isotope-delta values are commonly expressed in parts per thousand (per mil or ‰).

Accuracy was assessed by the repeated measurements of secondary reference material (RM) IAEA-600 (caffeine) and internal laboratory reference material (flour) for carbon and nitrogen stable isotope measurements and VSMOW 2, GISP, or SLAP 2 (all secondary RMs from IAEA) as well as laboratory reference material (tap water) for oxygen stable isotope analysis. Laboratory RMs were calibrated against secondary RMs by repeated measurements. Precision for all isotope ratio analyses was equal or better than 0.15‰.

Stable isotope values between warm and cold seasons were statistically compared using paired *t*-test (δ^15^N, δ^13^C in milk, and δ^18^O in precipitation) and Wilcoxon signed rank test (δ^18^O in precipitation). Normality was assessed using Shapiro–Wilk tests. For this study, *p* < 0.05 was considered statistically significant. All statistical analyses were performed using program R.

## 3. Results

The δ^18^O values (average ± SD) of milk water, drinking water (artesian), and precipitation are presented in Appendix A, Table A1. We found that the oxygen stable isotope ratio of artesian water in the Vilnius region was quite constant, and within a year, it changed little (−10.3 ± 0.3‰), whereas the precipitation oxygen stable isotope ratio values varied within ~20‰ range depending on the season (Figure 2). There were statistically significant differences in δ^18^O values between warm and cold seasons in precipitation (paired *t*-test, *p* < 0.05). Also, there were statistically significant differences in δ^18^O values between warm and cold seasons in cow milk water (Wilcoxon signed rank test, *p* < 0.05).

The oxygen stable isotope ratio in cow milk depends on the sources of drinking water as well as metabolism and the fractionation effect in milk production. The comparison of cow milk water, drinking water, and precipitation oxygen stable isotope ratios showed that milk water δ^18^O values were higher than the groundwater values by 1–8‰, depending on the season. The smallest difference between groundwater and milk occurred during the winter and transitional seasons, whereas the biggest differences occurred in the summer seasons. This reflects the change in water source during the year. In the colder seasons, the main source of water is groundwater, whose oxygen stable isotope ratio is rather constant, while in the warm season, an additional source of water—fresh grass that is enriched in ^18^O—is introduced to the cow diets.

The δ^15^N values in the cow milk varied from 2.9‰ to 6.0‰ (Figure 3) and did not differ statistically between cold and warm seasons (paired *t*-test, *p* = 0.07). These values for δ^15^N in the animal tissues are governed by the food nitrogen stable isotope signatures, in our case, grass and hay nitrogen stable isotope ratio values. In our study, plant δ^15^N varied from −1.8‰ (clover) to 4.3‰ (grass; see Appendix A, Table A2). Nitrogen-fixing plants have a similar stable isotope ratio to atmospheric nitrogen (~0‰), whereas plants absorbing nitrogen in the form of nitrates have higher δ^15^N values that are generally correlated with the nitrate source stable isotope ratio. Organic fertilizers have been found to have high nitrogen stable isotope ratio values, even exceeding 20‰ in some specific environments [24]. 

The δ^13^C values in the cow milk are presented in Figure 4; they varied from −31.2‰ to −27.6‰ and a statistically significant difference was observed between cold and warm seasons (paired *t*-test, *p* = 0.02). Cow milk had slightly lower carbon stable isotope values in the summer season, which was related to the higher ingestion of fresh grass, which is generally depleted in ^13^C. The carbon stable isotope ratio of the grass taken during the study was about −30‰, while the dried hay reached about −27.8‰. Generally, both higher and lower δ^13^C values indicate that these plants are within the C_3_ plant range, with an average of about −26‰. Herbaceous plants with a higher ^13^C fractionation usually fall under the lower values. Overall, the ratio of nitrogen stable isotope values did not change significantly in different seasons, while the ratio of carbon stable isotopes reflected the δ^13^C values of the feed source (fresh grass in summer and dried hay in autumn and winter) during the investigation period.

## 4. Discussion

There are three main sources of ingested water in herbivores: drinking water, water in plants, and atmospheric O_2_. The δ^18^O value of drinking water is related to the value of meteoric precipitation, which is correlated to mean annual temperature [26]. Plant water is enriched in ^18^O relative to groundwater via evapotranspiration. Enrichment is intense in arid areas [34] and its effects have been detected in the oxygen stable isotope analyses of mammals [35,36]. The turnover of oxygen and hydrogen stable isotopes in the body water, CO_2_, hair, and enamel has been analyzed in small mammals. A mass balance model on captive woodrats estimated that drinking water, atmospheric O_2_, and food were responsible for 56%, 30%, and 15% of the oxygen in the body water, respectively [37]. 

Models of isoscapes provide a means of interpreting observed isotopic data in terms of spatial patterns within the Earth. These models can help to predict patterns of environmental isotope variation that can be used to “fingerprint” the origin of geological and biological material. Isotopic signatures taken up by animal body tissues can similarly be related to location of origin [38]. Therefore, based on isotope distribution models, any unknown sample could be compared to a reference database in order to establish the likelihood of its geographical provenance [39]. This provides useful tools in verifying the declared origin of foods [40] and beverages [39]. According to the European annual map of oxygen stable isotope ratios in precipitation, the Lithuanian region falls within the range of −7.5‰ to −11.4‰ (Figure 5). That basically means that animal body tissues with a similar stable isotope ratio originate from that territory or are produced using water resources originating from within that territory.

In our study, oxygen stable isotope ratio measurements in milk samples were compared with precipitation and artesian water measurements as well as a European δ^18^O map in order to better understand the distribution of stable isotopes in cow milk. The ^18^O content of milk water should reflect the ^18^O content of the groundwater at the location where the cattle live because this is mainly what they drink. Thus, the δ^18^O measurement of milk water should also provide valuable information about the region of origin of the milk. The δ^18^O measurement of artesian water in our study area (Vilnius region) was −10.3 ± 0.3‰, which is comparable with the value reported by Mažeika et al. [41], who stated that the oxygen stable isotope ratio results of aquifers in the northeastern part of Lithuania range from −12.1‰ to −10.0‰. The δ^18^O values of paired milk and cow drinking water samples were strongly related in a predictable manner, as shown in Figure 2. As we expected, the stable isotope ratios of water within cow milk were comparable with geographical information (i.e., the stable isotope ratios of local drinking water) in cold seasons, when the main water source was artesian water (a metabolically driven stable isotope shift was expected compared with drinking water). However, the results of the measurements showed that stable isotope values varied over the course of the year in different seasons. During the winter/transition period, the δ^18^O values reflected the groundwater signal, while during the summer season, as a large part of the water comes from fresh grass, milk water tended to be enriched in ^18^O compared to groundwater. This is explained by the fact that a large amount of water is absorbed by fresh grass containing more ^18^O isotopes compared to groundwater due to the transpiration process, during which plants tend to remove the lighter isotope. During the summer period, the animals also lose more water, which results in the loss of the lighter isotope, whereas the remaining body water is enriched in ^18^O. 

Naturally occurring fertilizers usually have relatively high nitrogen stable isotope values (up to +30‰) [24], as volatile compounds tend to be more depleted in ^15^N, whereas solid compounds are more enriched in ^15^N [42]. Synthetic fertilizers are characterized by much lower δ^15^N values, ranging from −4‰ to 4‰ [43]. Thus, the nitrogen stable isotope ratio can be an indicator of farming strategy. In our study, the δ^15^N values in the cow milk varied from 2.9‰ to 6.0‰ and plant δ^15^N varied from −1.9‰ to 5.5‰. As the trophic fractionation of nitrogen stable isotopes in animal tissues is about 3‰, cow milk δ^15^N values indicate that the main nitrogen source for the cow was nitrogen from the grass (or hay). 

Different inputs of C_3_ and C_4_ plants in the diet can change the δ^13^C value in milk. Camin et al. [3] estimated that each 10% increase in the C_4_ plant content changes the δ^13^C value of casein by about 1‰. Skipitytė et al. [44] showed that this signal is reflected but with different levels of trophic fractionation in different tissues like muscles, feathers, skin, and blood. Cow milk is a product of animal metabolism and reflects the combination of both nutritional sources and processes in the body. In our study, we found that the diet of the cow consisted of C_3_ plants with δ^13^C ranging from −27.8‰ to −31.2‰, with slight changes according to the season.

## 5. Conclusions

Cow milk, drinking water, feed, and precipitation water were collected for three years from the same location. This allowed us to determine seasonal stable isotope dependences between cow milk water and drinking water. We observed that δ^18^O values were relatively higher during the summer season, whereas in the autumn, winter, and spring, they were relatively lower. Groundwater measurements indicated that the annual ratio of groundwater isotopes was relatively constant and had a δ^18^O value of −10.3 ± 0.3‰. Comparing our data obtained from European isotope maps, we found that only in the winter, spring, and autumn seasons does the ratio of oxygen isotopes of cow milk water correspond to the annual distribution of the European stable isotope ratio in precipitation. During the summer season, these values in the cow milk water are relatively higher and should be compared with the oxygen stable isotope distribution during summer months. By comparing the results of the summer season with the annual distribution, it is possible to draw conclusions about the origin of dairy products. Therefore, the analysis of seasonal isotope fluctuations is important in improving the methods available for identifying the origin of various foods.

We found that the carbon stable isotope ratios were correlated with the contribution of dietary sources (fresh grass in summer and dried hay in winter), meanwhile nitrogen stable isotope ratios did not differ significantly during the year. Overall, combined δ^18^O, δ^13^C, and δ^15^N analysis of cow milk provides valuable information about the origin of milk and the dietary regime of animals and can be helpful for studies of milk product authentication. However, our analysis showed that seasonal variations in isotopic ratios are typical in natural samples. Thus, databases and comparative material are needed to determine the origin of food products.

## Figures and Tables

**Figure 1 animals-09-00069-f001:**
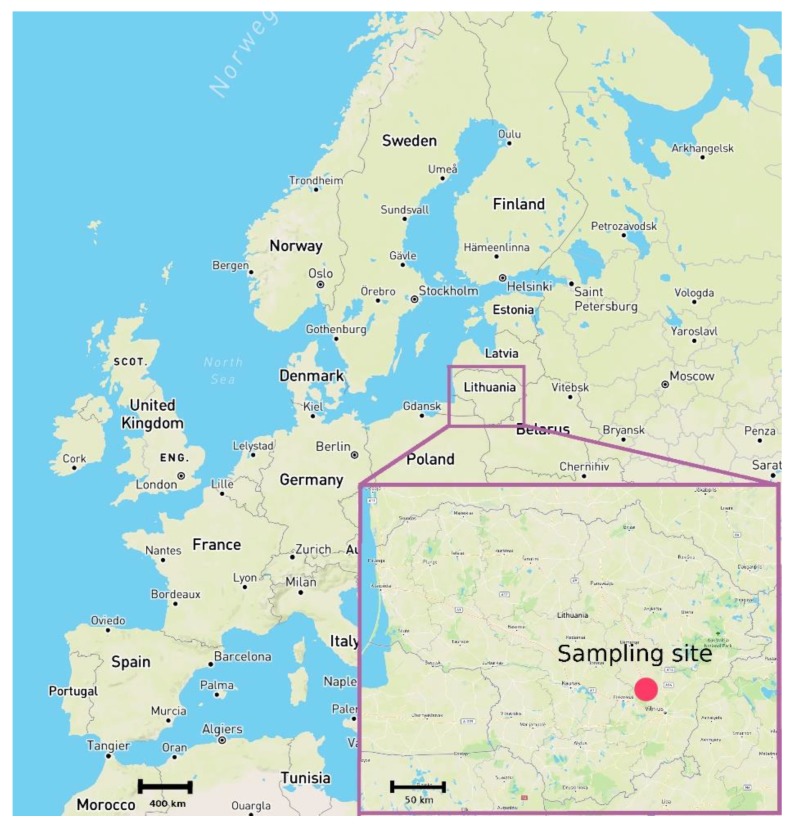
Map of Europe with the location of the farm (marked as red dot in the inset) in Lithuania.

**Figure 2 animals-09-00069-f002:**
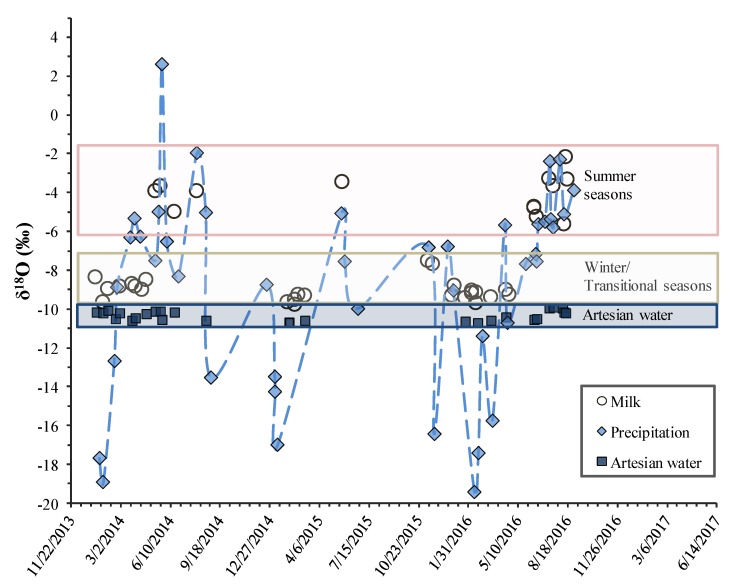
Seasonal variation in δ^18^O values in milk water, artesian water, and precipitation water in the rural site in Lithuania during 2014–2016. Dashed blue line integrates measured δ^18^O values of precipitation. Boxes indicate which δ^18^O values of the cow milk fall into the range of summer or winter/transitional seasons.

**Figure 3 animals-09-00069-f003:**
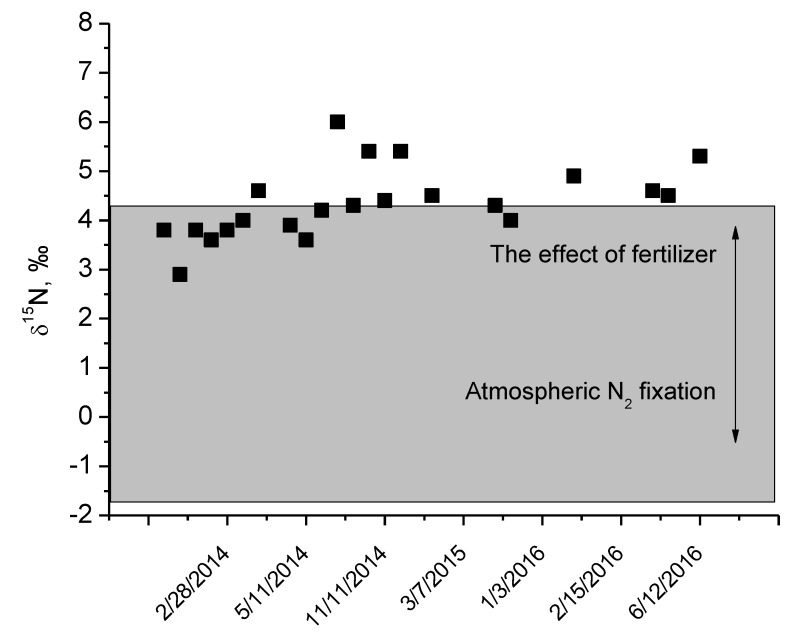
δ^15^N values in cow milk (squares). The grey shaded area represents the measured δ^15^N values of the grass.

**Figure 4 animals-09-00069-f004:**
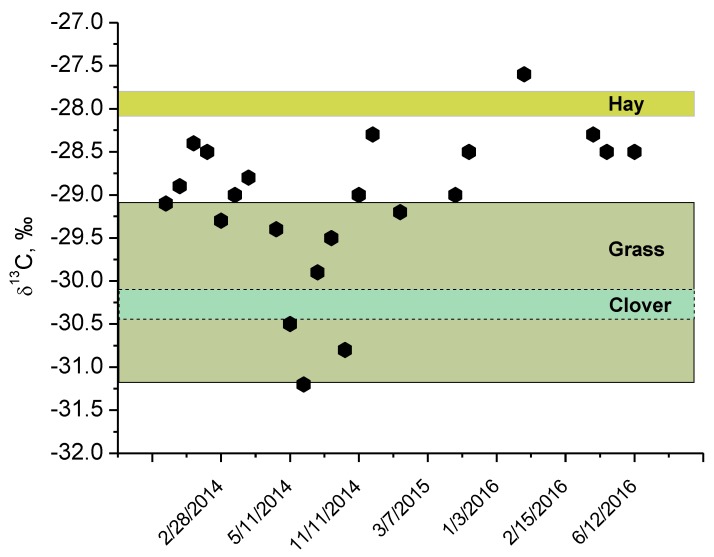
δ^13^C values in the cow milk (dots). The boxes represent the measured δ^13^C values for the hay, grass, and clover. The measured δ^13^C values of clover are overlapping with the δ^13^C values of the grass.

**Figure 5 animals-09-00069-f005:**
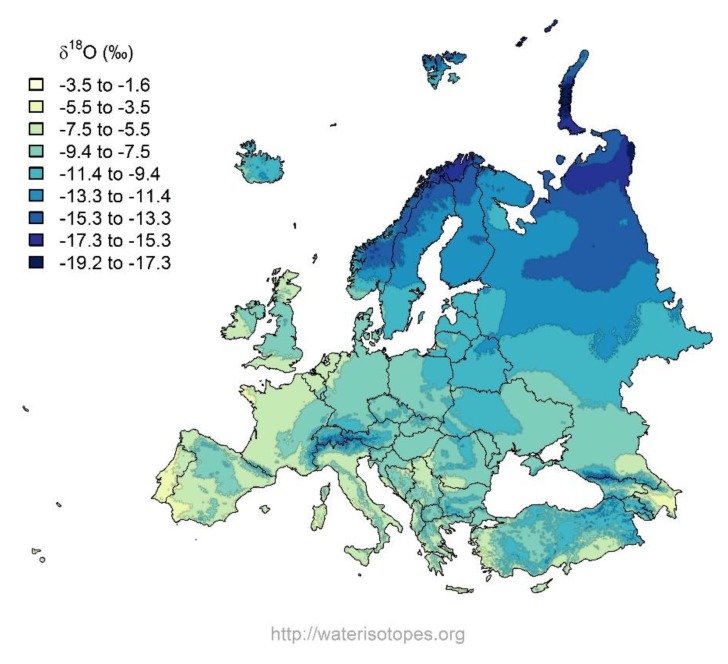
Distribution of oxygen stable isotope ratio in the precipitation in Europe [27].

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
