# Peer review of "Seasonal Variation in Stable Isotope Ratios of Cow Milk in Vilnius Region, Lithuania"

_animals, 2019, doi:10.3390/ani9030069_

Round 1
Reviewer 1 Report
I have now read the paper " Seasonal variation in stable isotope ratios of cow milk ". The authors present a paper that aimed to investigate how the isotope composition of milk varies during the seasons. The manuscript shows a good idea of research, the results look reasonable. However, in spite this data are really useful, some parts the manuscript need to be revise and some concepts have to be improve. Lots of information about sample procedures are not reported (sampling dates, number of samples…).
The manuscript can be published after major comments (in attachment).

Author Response
Response to Reviewer 1 Comments
Review of the manuscript: Seasonal variation in stable isotope ratios of cow milk I have now read the paper " Seasonal variation in stable isotope ratios of cow milk ". The authors present a paper that aimed to investigate how the isotope composition of milk varies during the seasons.
The manuscript shows a good idea of research, the results look reasonable and is certainly within the scope of the journal.
In particular, these data improve the gap of knowledge in Lithuania area that is really poor in terms of isotopic information. However, in spite this data are really useful, some parts the manuscript need to be revise and some concepts have to be improve. Lots of information about sample procedures are not reported (sampling dates, number of samples…).
The manuscript can be published after major comments (in attachment).
Title: In the title is good to report the study area.
Response: The title was modified.
Introduction: need to be revise and to add information about other recent study in this field (i.e. isotopic studies in Milk in different region, see the references add at the end of this text).
Response: The introduction was updated with recent literature citations.
Please revise the Materials and methods chapter, in particular:
Line 79: the samples collection is not so clear. Is better to add a table that reports n. samples, dates, samples type.
Please add also information about the milk, for examples if you collect milk samples from the same cow or not.
Response: The tables were added in appendix. All measurement points are presented in these tables.
Line 85: Fig. 1 – Is it possible to increase the resolution of the Fig. 1 - inset? It would be nice, also to report Fig. 1a (Europe) and Fig.b (the inset). What the green dot means? I think that the arrows indicate the farm, right? It would be nice, also to have a scale, to have an idea of the distance, in the inset.
Response: Fig. 1. was modified. Resolution improved, notes added.
Line 91: In this study I understood that you collect precipitation during rainfall events, is correct? Consider that, when you measure isotopic values during a RE is possible to have really different results (Tiang et al., 2018 - Scientific Reports volume 8, Article number: 6712). Also, in this case, it’s really important to take into account also the total rainfall during the event (mm). For example, you can have d18O=xxx and dD=yyy but the event was only 10 mm. That is not significant (or we don't know) for a correct study, in this case.
In these studies would be better to collect monthly precipitation, you can use a water tank, filled with paraffin (IAEA/GNIP Precipitation sampling guide) and then consider the monthly precipitation in a station close to your study site.
Response: We collected samples after the raining events. All water from one rainfall event was collected in the vessel, and part of the water was taken to the analysis. During the cold season precipitation was collected periodically. During the warm season precipitation was collected immediately to avoid evaporation. Very short-term events were not included.
Line 138: Table 1 – Do the values represent and average of (example): spring 2014 - spring 2015 - spring 2016...etc? Please consider the comments in the line 79. Did you take water sample from precipitation in different days/weeks from the wells and the milk? This is not clear.
Response: The tables were added in appendix.
Line 142: Fig. 2 – Need important revisions. The blue line for precipitation indicates that you collect rainfall in which days? I think you didn’t measure d18O in continuous. In this case the representation is not correct. Please add a symbol for precipitation (squares, dots…) and use the broken line in spite of the round (chamfer) line.The blu bar for the Artesian water is an average value (Table 1) but is represented as a continuous line for the entire period. Is it possible to consider isotopic value stable for the three years? Also, you never mentioned the depth of the well. It's better to change the colour of the precipitation line or the colour of the blue bar for the well (the same colour may confuses the reader). In the x-axis: the interval of the date is not clear, is not possible to understand when you sample the milk, the precipitation… because you didn't represent all the dates and in this way is not useful. What the date represents (Precipitation)? It’s really important to add symbols and broken line, this is formally not correct.
Response: Fig. 2 was improved according to the reviewer suggestions
Consider recent literature review, important to understand better how to present data and to have an updated idea of similar research in other countries:
Ehtesham et al. (2013) Correlation of Geographical Location with Stable Isotope Values of Hydrogen and Carbon of Fatty Acids from New Zealand Milk and Bulk Milk Powder. Agric. Food Chem., 61 (37), pp 8914–8923. DOI: 10.1021/jf4024883.
Chesson et al., (2018) Hydrogen and Oxygen Stable Isotope Ratios of Milk in the United States. Journal of Agricultural and Food Chemistry 58(4):2358-63
James F. Carter, Lesley A. Chesson (2017) Food Forensics: Stable Isotopes as a Guide to
Authenticity and Origin – Book.
Response: Literature citations were added to the manuscript.

Reviewer 2 Report
see the attached file

Author Response
Response to Reviewer 2 Comments
The paper is interesting and quite innovative. I found interesting in particular the relationship between milk d18O and precipitation d18O. Differently from previous paper, it results that the increase of d18O in summer is not due only to the contribution of fresh forage in the animal diet, but there is a significant link also with precipitation d18O.
As for d15N and d13C, I do not find here a significant seasonal variation, but authors in abstract and conclusion state differently. This aspect should be reviewed. I also suggest authors to apply statistical analysis to the data, in order to better understand these variations and relationship.
Response: statistical analysis was added to the manuscript. From these analysis we can more strongly state that there are statistically significant seasonal differences in milk δ13C values, and there is no statistically significant differences in milk δ15N values in warm and cold seasons of the year.
As for English form, it needs in some points a revision by a native English speaker.
Response: Manuscript was edited by professional English editing service by MDPI.
In details:
Row 13 and 25: replace 18O with d18O, or 18O/16O
Response: Replaced.
Row 28: the correct way of writing is ‘oxygen stable isotope ratio’ and not ‘stable oxygen isotope ratios’, because stable refers to isotope not to oxygen. Review it also in the rest of the text.
Response: Edited.
Row 37: analytical instead of bioanalytical is more correct.
Response: Changed.
Rows 44-46: more appropriated references on lamb, or chicken or review on animal products are missing. Some examples:
The review Camin, F., Bontempo, L., Perini, M., Piasentier, E. Stable Isotope Ratio Analysis for Assessing the Authenticity of Food of Animal Origin (2016) Comprehensive Reviews in Food Science and Food Safety, 15(5), pp. 868-877. And the reference inside.
Response: Additional references were added.
Row 50: stable isotope ratio of nitrogen (expressed as d15N) instead of ‘nitrogen’. The same above for C and then for O. In this way, you can afterword use the delta notation when you refer to the isotopic ratios.
Response: Edited according to the reviewer suggestions.
Row 59: three stable isotope ratios, instead of isotopes (specify the ratio into the brackets: e.g. 13C/12C…)
Response: Edited according to the reviewer suggestions.
Rows 60: some stable isotope ratios or some isotopic ratios (15N/14N, 34S/32S)
Response: Edited according to the reviewer suggestions.
Row 68: I would deleted ‘hydrogen’, because you have not written about it previously
Response: Deleted “hydrogen”.
Rows 71-72: on d18O, d13C and d15N
Response: Edited according to the reviewer suggestions.
Rows 104-105: in Scampicchio et al., 2012, as well in many other papers on stable isotope ratio analysis of milk, the d13C and d15N analyses have been performed on the fractions of milk (casein, whey and fat) and not on bulk freeze dried milk. This is due to the fact that fat has very different d13C than casein. Authors have to explain the reason why they decided to analysis milk, instead of the single fraction.
Response: Bulk milk powder measurements are not so precise, though also informative and predictable way to investigate the whole diet. Camin et al (2008) revealed that δ13C values of bulk milk were between those of casein and lipids. Deviations can be found due to different digestibility and therefore preferential assimilation of a certain nutrient by the animals. In the same study it was found that δ15N values of bulk milk were about 0.4‰ lower than those of casein. The other study (Ehtesham et al, 2013) showed that δ2H and δ13C of bulk milk and several fatty acids correlated with regional production area. It was also mentioned that conventional analytical methods that use stable isotope data to determine origin measure the isotopic composition of the bulk sample and hence are applicable to pure (single-source – as it is in our study) products but are unable to distinguish the origin of adulterated dairy products once they are incorporated into mixtures. Knobbe et al. (2006) and Kornexl et al. (1997) confirmed that the d C of bulk milk, as well as of casein and glycerol extracted from milk or cheese, is related to the amount of C3 and C4 plants in animal diet. Therefore, both bulk and fractional analyses can be used depending on the aims of the studies. Since the main focus was on oxygen stable isotope ratio variation, δ13C and δ15N measurements gave additional information on animal diet.
Literature:
Camin, F., Perini, M., Colombari, G., Bontempo, L., & Versini, G. (2008). Influence of dietary composition on the carbon, nitrogen, oxygen and hydrogen stable isotope ratios of milk. Rapid Communications in Mass Spectrometry: An International Journal Devoted to the Rapid Dissemination of Up‐to‐the‐Minute Research in Mass Spectrometry, 22(11), 1690-1696.
Ehtesham, E., Hayman, A. R., McComb, K. A., Van Hale, R., & Frew, R. D. (2013). Correlation of geographical location with stable isotope values of hydrogen and carbon of fatty acids from New Zealand milk and bulk milk powder. Journal of agricultural and food chemistry, 61(37), 8914-8923.
Knobbe, N., Vogl, J., Pritzkow, W., Panne, U., Fry, H., Lochotzke, H. M., & Preiss-Weigert, A. (2006). C and N stable isotope variation in urine and milk of cattle depending on the diet. Analytical and Bioanalytical Chemistry, 386(1), 104-108.
Kornexl, B. E., Werner, T., Roßmann, A., & Schmidt, H. L. (1997). Measurement of stable isotope abundances in milk and milk ingredients—a possible tool for origin assignment and quality control. Zeitschrift für Lebensmitteluntersuchung und-Forschung A, 205(1), 19-24.
Rows 120-126: The expression of the results does not comply with the IUPAC requirement:
According to Brand et al., 2014, the delta notation is calculated according to the following general equation (without x1000):
δi E = (i RSA − i RREF )
i RREF
where i is the mass number of the heavier isotope of element E (for example, 13C); RSA is the respective isotope ratio of a sample (such as for C: number of 13C atoms/number of 12C atoms or as approximation 13C/12C);
RREF is the relevant internationally recognized reference material.
Then you should write that ‘The delta values are multiplied by 1000 and are expressed in units “per mil”(‰).’ Moreover you have to describe which standards (I guess they are working standards) you have used for calibration/normalization of the results, and how you have done calibration of the working standards.
Response: The formula was corrected. Additional data on the secondary reference materials and calibration was added.
Row 133: I would replace ‘stable oxygen isotope ratio values’ with ‘d18O values’. Anyway the correct way of writing would be ‘oxygen stable isotope ratios’, because stable refers to isotope not to oxygen.
Response: Edited according to the reviewer suggestions.
Rows 154-161: this paragraph is not clear and must be rewritten.
Response: Rewritten, moved to Discussion.
Row 165: the values actually are not so low, but comparable with others in the literature.
Response: Deleted “low”, as they are comparable with others in the literature.
Row 179: have you mathematically computed this relationship, checking its statistically significance?
Response: Statistical analysis was performed and added in text above (both for nitrogen and carbon stable isotope ratio data).
Rows 209-210: ‘which in turn..seasons’: delete this sentence
Response: Deleted.
Row 216: I do not understand the meaning of ‘biogens’.
Response: Deleted.
Rows 221-222: this sentence is not clear and needs rewriting.
Response: Rewritten.
Rows 228-229: I agree that d13C slightly changed, but at rows 178-181 and 244-245 you have written the contrary. It is necessary to homogenize.
Response: Clarified. Statistical analysis showed a slight but statistically significant difference between warm and cold seasons for carbon stable isotope values, meanwhile there was not found statistically significant differences in nitrogen stable isotope ratios between the seasons.
Row 247: about the origin of milk and the dietary regime of animal.
Response: Changed.

Reviewer 3 Report
The proposed article is quite interesting.
Here are some questions and proposals for authors
1) Why you did not use statistical tools for you results?
2) It would be nice to present all your results for each elemental ratio numerically in supplementary material.
3) Figures 1 & 2 you can delete the frameworks.
Line 45: You can use also more recent references
Line 51: What do you mean differentiate between trophic levels?
Lines 53-54: Please make a small refer to which plants follow the photosynthetic pathway C3, C4 and CAM
Line 62: It would be nice to delete “properties” and write “indicators” OR “markers”
Line 74: You could delete “the”
Line 92: The degree is underlined, please correct.
Line 102: “et al.” Please do not use italic
Line 172: “The grey box represents the measured δ15N values of the grass” Which is the mean and the SD of δ15N values of the grass?
Line 225: “et al.” Please do not use italic
Author Response
Response to Reviewer 3 Comments
Comments and Suggestions for Authors
The proposed article is quite interesting.
Here are some questions and proposals for authors
1) Why you did not use statistical tools for you results?
Response: We added statistical analysis in the revised manuscript.
2) It would be nice to present all your results for each elemental ratio numerically in supplementary material.
Response: The tables were added in appendix.
3) Figures 1 & 2 you can delete the frameworks.
Response: Deleted
Line 45: You can use also more recent references
Response: More recent references were added.
Line 51: What do you mean differentiate between trophic levels?
Response: We added “in animals”. By saying “differentiate” we meant to estimate trophic position of the animal.
Lines 53-54: Please make a small refer to which plants follow the photosynthetic pathway C3, C4and CAM.
Response: Examples of plants were added.
Line 62: It would be nice to delete “properties” and write “indicators” OR “markers”.
Response: Changed.
Line 74: You could delete “the”
Response: Deleted.
Line 92: The degree is underlined, please correct.
Response: Corrected.
Line 102: “et al.” Please do not use italic
Response: Italic changed to regular.
Line 172: “The grey box represents the measured δ15N values of the grass” Which is the mean and the SD of δ15N values of the grass?
Response: The mean and the SD of δ15N values of the grass are presented in Appendix A. We measured additional samples of different types of cow feed, and added new results to the Fig. 3.
Line 225: “et al.” Please do not use italic
Response: Italic changed to regular.

Round 2
Reviewer 1 Report
Overall comments:
This is an interesting paper about the isotopic signature of milk in Lithuania region. It is a pleasure to me to read the manuscript and I hope that you appreciate the comments. After the revision, the paper clearly merits publication in Animals.
Reviewer 3 Report
Line 52: please delete the comma after CAM
Line 83: Please delete "the"